# Mitigating the Risk of Autonomous Weapon Misuse by Insurgent Groups

Jonathan Kwik

Faculty of Law, University of Amsterdam, 1018WB Amsterdam, The Netherlands; h.c.j.kwik@uva.nl

**Abstract:** The intersection between autonomous weapon systems ('AWS') and non-State armed groups ('NSAG') is an underexplored aspect of the AWS debate. This article explores the main ways future policymakers can reduce the risk of NSAGs committing violations of the laws of armed conflict ('LOAC') using AWS once the technology becomes more prolific and easily distributable. It does this by sketching a chronological picture of an NSAG's weapons obtention process, looking first at its likely suppliers and transport routes (acquisition), and, subsequently, at factors which can increase the risk of LOAC violations once the system is in their possession (use). With regard to use, we find that the lack of explicit legal obligations in LOAC to (a) review weapons meant solely for transfer and (b) provide technical training to recipients of transfer constitute serious reasons why LOAC violations may be aggravated with the introduction of AWS to insurgent groups. We also find, however, that States are uniquely and powerfully placed to address both acquisition and use factors, and outline how they can be persuaded into implementing the risk-reducing measures recommended in this article for purely strategic reasons, i.e., even if they express no interest in improving LOAC compliance per se.

**Keywords:** autonomous weapon system; artificial intelligence; insurgency; armed group; IHL; LOAC; arms control

## 1. Introduction

In 2017, the YouTube channel (Stop Autonomous Weapons 2017) published a video titled *Slaughterbots*, depicting a disconcerting projection of what AI and robotics technologies could lead to. In the short film, a company is shown promoting small swarming drones equipped with face recognition and an explosive payload, designed to infiltrate buildings and kill persons with a matching signature. To much applause of the audience, the drones are proudly touted as "precise" and "cheap", making many conventional weapons obsolete. Predictably, nefarious actors are quickly shown taking advantage of these products, using them to carry out targeted political and ethnic killings. In the penultimate scene, when asked who could be behind all those attacks, an interviewed analyst responds dryly: "Could be anyone".

This film was produced in the context of a broader international campaign against the development of autonomous weapon systems ('AWS'), novel technologies which leverage the capabilities of modern artificial intelligence ('AI') to improve warfighting capacity.[1] Starting as a campaign for placing limits on robotic military hardware in 2009 (Fleming 2009), AWS quickly gained international attention from NGOs such as (Human Rights Watch 2012), the UN (Heyns 2013), the EU (European Parliament 2014, 2018, 2020), the (ICRC 2013), and eventually was taken up by the Convention on Certain Conventional Weapons Conference (GGE LAWS 2019; Prigg 2016). For a majority of these debates, the role of the State has remained a core presumption: popular topics of discussion include

---

[1] These are variably referred to in discussions as 'lethal autonomous robots', 'fully autonomous weapons', and 'killer robots'.

whether (inter-State) prohibitory or regulatory treaties should be adopted, how States can ensure ethical and legally compliant development and use, and how States can guarantee accountability for the use of such weapons.[2] In contrast, one of the core messages of the *Slaughterbots* film—the threat of distribution to third, non-State parties—has received little scholarly attention. To the author's knowledge, there is minimal literature addressing the intersection between AWS technology and NSAG weapon acquisition dynamics.

This lacuna is troublesome as a majority of contemporary conflicts involve armed groups,[3] making it crucial to analyse the impact of AWS at the sub-State level. In light of this, in this article, we discuss the potential dangers of the development, transfer and use of AWS by moving our focus away from the State and toward organised non-State armed groups ('NSAG'). More specifically, we discuss how these weapons may affect legal compliance to the law of armed conflict ('LOAC') in internal struggles, such as insurgencies and rebellions. Generally speaking, insurgencies take longer, are messier, and involve more LOAC violations in comparison to conventional State-against-State wars (Bakke 2013),[4] potentially making AWS a further aggravating factor. A better understanding of this relatively overlooked category of potential users will be to the benefit of the overall AWS debate and will enable stakeholders, governments and inter-State organisations to make more timely and informed policy decisions both at the national and international level.

The current paper engages in a preliminary examination of this crossroads between AWS technology and insurgent groups. Owing to the limited existing scholarship in this field, the aim of this article is primarily exploratory: to identify core dynamics and challenges related to these topics and spark further discussion. The analysis in this article draws from an extended literature review of existing general theories concerning weapons transfers to NSAGs, and such groups' violation propensities. From these baselines, we then develop prognostics on how these dynamics might change in an age where AWS are more prolific. As it is difficult to predict precisely what form autonomous technologies will take in the future (Boddens Hosang 2021; Schuller 2017), most of our findings will take the form of informed hypotheses, drawing from existing systems and current trends,[5] which, nevertheless, can be used to advise future debate, diplomacy and policy.

This article proceeds as follows. In Section 2, we briefly discuss the technology in question and certain technical aspects that can potentially impact legal compliance by NSAGs, and why the focus on NSAGs is important. Then, we discuss the potential impact toward LOAC compliance from two distinct, but not entirely unrelated, perspectives. First, in Section 3, we discuss the potential ability of NSAGs to *access* the technology and the possible routes through which they can acquire it. This is an important consideration, as NSAGs evidently cannot misuse technology which is not in their possession, and policy can be adopted targeting groups' ability to secure such systems. Second, in Section 4, we look at the situation where NSAGs already are in possession of such technologies: here, we explore different factors that can potentially lead NSAGs to *use* the technology unlawfully. In this section, we start from more general theory concerning the different causes of LOAC violations by insurgents, before moving to more specific analysis of how AWS-specific characteristics can amplify (or reduce) this tendency. We identify four general ways NSAGs can misuse AWS technology: deliberately, due to organisational factors, because the system itself is problematic, and due to difficulties in understanding the technology. As with the acquisition phase, targeted policy decisions (such as technical sensibilisation efforts) can be made to address each factor once they have been more clearly understood. We conclude, in Sections 5 and 6, with some recommendations for States, stakeholders and the international

---

2　For summaries of the overall debate and the constituent points of contention, see (Cummings 2018; Eklund 2020; Lewis 2015; Santoni de Sio and van den Hoven 2018).

3　See Section 2.2.

4　This does not take away from the fact that some international armed conflicts can also be particularly violation-dense, as observed during the Yugoslav wars and the 2022 Russo-Ukrainian war. However, existing literature on AWS plentifully addresses this type of conflict.

5　See Section 3.2.

community to mitigate the risks identified in this paper. Intriguingly, we find that States can potentially be persuaded into adopting several of the recommended measures purely for strategic reasons, i.e., even if the State is not particularly interested in improving LOAC compliance per se. This increases the action space available for groups aiming to lobby for such measures to be implemented, such as NGOs and the ICRC.

For clarity, when we discuss AWS in the context of this paper, we are referring to weapon systems which go beyond simple handmade drones or mechanisms; rather, we are referring to more sophisticated forms of modern autonomous systems which require some investment and production capacity to manufacture, such as autonomous armed drones (Boddens Hosang 2021), swarms (Safi 2019), active protection systems (Scharre and Horowitz 2015), or loitering munitions (Gettinger and Arthur 2017). We are, therefore, taking a slightly narrower view than some popular definitions of AWS, such as that provided by ICRC.[6] The reason for this is that an overly broad definition could potentially include any rudimentary mechanism that autonomously fires a shot whenever its AI notices some movement through its camera (Scharre 2018). Such means of warfare, while technically also 'autonomous' under certain definitions (e.g., (Sartor and Omicini 2016; Scharre and Horowitz 2015)), are excluded from this discussion for the same reason boobytraps and tripwires are excluded from arms-control and arms-trade studies: they are practically impossible to trace and to collect data for in a field already struggling with transparency and data collection (Smith and Tasiran 2005). As such, we will restrict the analysis of this article to weapon types and weapon systems generally covered in arms-trade research, but which are enhanced by AI.

With regard to the armed groups themselves, there is no standardised definition of what constitutes an NSAG, and there are many alternate terms found in literature ('armed non-State actor', 'armed opposition group', 'insurgent'). For the purposes of this article, they are defined broadly as organised non-governmental groups who directly or indirectly engage in support of non-governmental combatants (Bassiouni 2008). Notably, a legal distinction must be made with armed groups fighting against colonial domination, alien occupation and racist regimes. While such groups might functionally be indistinguishable from 'standard' NSAGs, they legally trigger a completely distinct framework of rules under LOAC (Additional Protocol I 1977) which will not be addressed in this paper. This analysis will limit itself to legal norms applicable to non-international armed conflicts ('NIAC')—conflicts geographically restricted to a single State between that State and one or more NSAGs, or between two or more NSAGs (ICRC 2014). To maintain generalisability, we assume that the conditions required for their classification as a party to the conflict (Bartels 2018; Geneva Convention IV 1949), such as the existence of a NIAC and sufficient organisation, are satisfied.[7] Groups too disorganised or weak to provide some measure of concerted or sustained opposition are not considered, as these fall outside of the LOAC framework (Additional Protocol II 1977; Bassiouni 2008).

## 2. Background

### 2.1. How AI Transforms How Weapons Work

AI is a universal field which lies at the heart of many conveniences we take for granted (Russell and Norvig 2010), and has become the focus of international competition (Boulanin et al. 2020). Almost all world powers have explicitly emphasised the importance of developing AI and are investing heavily in the technology (Dailymail 2017; Gao 2017; Ministère des Armées France 2019; Roy 2018; Thorne 2020). The military domain is no different, and

---

[6] While the conception of the specific technology being addressed varies significantly by source, one popular definition by the ICRC (Davison 2017) is formulated as follows: "Any weapon system with autonomy in its critical functions—that is, a weapon system that can select (search for, detect, identify, track or select) and attack (use force against, neutralize, damage or destroy) targets without human intervention". For the lack of consensus on a definition in the international debate, see (Boulanin 2016).

[7] This does not a priori exclude groups labelled as 'terrorist organisations'—strictly speaking a political designation—as long as they are parties to the conflict. For a more complete discussion on when terrorist groups qualify as parties under LOAC, see (Bartels 2018).

AI has been cited as highly desirable for providing increased military efficiency (Heyns 2013), providing higher speed and precision compared to the human soldier (Defense Innovation Board 2019; Ministère des Armées France 2019), shortening the decision-making cycle (United States Air Force 2009), improved intelligence collection and analysis (DoD Defense Science Board 2016), and being able to function in communications-denied environments (SIPRI 2017). In combination with robotics, they allow force projection in an area without endangering friendly personnel (ICRC 2018a; Thurnher 2014). There are also practical incentives: advocates (Scharre 2011; Schmitt and Thurnher 2013; Schuller 2017) cite these systems' ease of production in addition to a reduction in costs and logistical burdens (vis-à-vis training and maintaining human soldiers).

Most of these new possibilities are attributable to the impressive developments in AI technology in the past decades. In particular, the power and speed of modern AI are made possible by a combination of two factors (Samek and Müller 2019). Firstly, AI are powered by increasingly powerful GPUs. Since the Second World War, there have been consistent improvements in speed and processing capacity over time coupled with decreases in cost (Russell and Norvig 2010), making better technology available at increasingly affordable prices. Secondly, machine-learning techniques have revolutionised the way AI can be programmed. In short, machine learning allows programmers to design AI based not on handcrafted rules but through patterns derived from data (Molnar 2019). Not only does machine learning allow AI to be programmed for situations which were previously impossible to define or model by hand (Brownlee 2020), but these models also boast impressive levels of performance. Deep neural nets, in particular, are regarded as both extremely accurate and very versatile (Independent High-Level Expert Group on Artificial Intelligence 2019). It is no wonder, then, that machine-learning models have become practically ubiquitous, being found at the core of many recent advances in science, robotics, and engineering (Ribeiro et al. 2016; Righetti 2016).

In weapon systems, AI can fulfil a variety of functions. These can range from operational functions such as mobility, health management, and communications, to more critical tasks such as information analysis and target selection (Boulanin 2016). Naturally, different AI can be combined into one system, enabling it to perform multiple functions simultaneously. In the policy debate, AI designed to make targeting decisions usually generate the most controversy—let us, thus, focus on the ramifications of this type of functionality. From the targeting perspective, opponents often argue that it is impossible for AWS to function within applicable legal constraints, either because the rules themselves are too complicated or subjective, or because the overall operating environment is (Garcia 2014; ICRC 2018a; Li and Xie 2019). We will not further address the accuracy of these arguments here, as this discussion is still ongoing and involves a rather complicated analysis of both the technical capabilities of AWS and the specific legal requirements in LOAC.[8] It must nevertheless be emphasised that irrespective of the current state of AI technology and whether the arguments are correct, there is universal consensus that the use of new weapon technologies must at all times respect LOAC rules in force (Jensen 2020; Schmitt and Thurnher 2013; Stürchler and Siegrist 2017). This unequivocal minimum legal standard in mind, we can proceed with our discussion while leaving evaluations of individual systems, with their particular specifications, for more casuistic, applied analyses.

### 2.2. The Relevance of NSAGs

(Heffes 2013) refers to NSAGs as "one of the most important actors within the international scene, particularly in [LOAC]". NSAGs are important to bring into the overall debate concerning AWS for one main reason: most modern conflicts involve them as active participants, and their actions have a significant impact on civilian lives, sometimes even more so than State actions. Since the Second World War, there has been an exponential

---

[8]　In addition, some are of the opinion that this argument cannot be resolved as it involves speculating how AI will develop in the future, which is not productive. See (Schuller 2017; UK Mission Geneva 2019; Wittes 2012).

rise in 'internal' or 'non-international' armed conflicts, and many featured more than one NSAG.[9] In the period of 1989–1997, only 6 of the 103 armed conflicts were between States (Blanton 1999), and from 1990–2001, 95% of conflicts were reported as internal (SIPRI 2002). Indeed, (Saul 2017) notes that "most contemporary armed conflicts are non-international, involving thousands of NSAGs worldwide that have the potential to both protect and endanger civilians". Often, these conflicts are irregular and asymmetrical, are waged in areas with civilian concentrations, and rarely feature conventional, 'set-piece' battles (Heyns 2013; Wood 2008). Many of these conflicts directly impact the civilian population, who are victimized by both NSAGs and State forces (Atadjanov 2011; Pejić 2011).

Because of the central role played by NSAGs, more emphasis has been placed on reaching out toward armed groups through sensibilisation and dissemination efforts (Bellal 2016; ICRC 2018b). A 2010 UN Resolution (UN Security Council 2010) recommended "systematic and consistent engagement with non-State armed groups" to improve the overall humanitarian situation during internal conflicts. Prominent LOAC jurist (Sassòli 2008), emphatically calling himself an "extremist on the issue", strongly promoted engaging all armed groups (even those one may want to label as 'terrorists') as a way to improve LOAC compliance. For many of these proponents, then, focusing on the NSAG level is as important as the State level when the ultimate goal is improved overall respect for the laws of war.

NSAGs are often also attributed a higher propensity to violate LOAC. While not necessarily applicable to all armed groups—many NSAGs are known to be disciplined and to show active commitment to upholding humanitarian norms (Saul 2017)—a significant body of literature[10] agrees that low compliance to LOAC by armed groups is a significant contributor to humanitarian harms during insurgencies. To add to this concern, globalisation and weak international regimes on arms trade have made it much easier for NSAGs to access increasingly sophisticated weaponry.[11] As weapon availability is determinant of an NSAG's ability to pursue violence (Jackson 2011), the overall humanitarian situation is directly impacted by the weapons available to active NSAGs within that conflict.

Let us now consider in more detail the likely scenarios leading to NSAGs acquiring AWS technology and know-how. Note that because we are speaking of AWS generally (and not specific systems), these conclusions will reflect that: there will always be particular systems which deviate from the norm and to which these findings will not apply. There is merit in maintaining this more general approach, however, since this paper aims to sketch an initial picture of the risks attached to AWS transfers as a whole; these assumptions can then be adjusted afterwards based on a particular weapon's more specific characteristics.

### 3. Acquisition Sources and Routes

*Transfer* is a general term used to describe the movement of weapons (systems) to or from a particular group or entity (Arms Trade Treaty 2013; Clapham et al. 2016). The term 'transfer' is preferred here over alternatives such as 'export' or 'sale' as these imply some form of economic activity. It is important not to limit our discussion to pure sale-and-purchase situations, as in practice, arms are also transferred in return for favours, loans, price reductions and barter deals, or simply as donations (Thurner et al. 2019). The Arms Trade Treaty commentary (Clapham et al. 2016) similarly emphasises that the term 'transfer' should include non-compensated exports (as otherwise, State parties can simply disguise all transfers as 'gifts' and evade their obligations). The actors involved do not have to be States, although it is common that at least one State is present (often as the donator or producer). The SIPRI database,[12] the leading source for arms-transfer studies (Moore 2012), clarifies that it suffices that the transfer is trans-group, i.e., between "one country, rebel

---

[9] For a graphical representation of the development of internal versus international armed conflicts, see (Harbom and Wallensteen 2005).

[10] See Section 4.

[11] See Section 3.

[12] See www.sipri.org/databases/armstransfers, accessed on 10 October 2022.

force or international organization" to another, with any combination being valid (SIPRI 2020). As such, for the purposes of this article, any type of arms obtention by NSAGs from an external source can be considered a 'transfer'.

Arms transfers are not necessarily restricted to the actual weapons themselves. Transfer encompasses both technology and knowledge. As (Cragin et al. 2007) explain, technology refers to physical tools and devices, while knowledge refers to know-how: "tactical plans, intelligence, and other information including the data and expertise needed to use specific technologies well". Know-how is just as important as the weapon itself: "For any given new technology to be effective, the receiving organization also must have the appropriate knowledge to use the technology successfully." (Jackson 2011) gives some examples of the Sudan People's Liberation Army, the Mozambican Renamo, and Angolan UNITA, where their lack of know-how prevented them from optimally using the rather powerful arms they had obtained.

Arms access and possession has been frequently linked with a worsened humanitarian situation in the field. For example, (Byman et al. 2001) found that "[arms] support for an insurgency can make a movement far more effective, prolong the war, increase the scale and lethality of its struggle, and may even transform a civil conflict into an international war". The (ICTY 1995) found that the frequency of civil wars is at least partly due to the fact that "technological progress has made it easier for groups of individuals to have access to weaponry". (Stewart 2017) directly links the widespread access to weapons and technology of NSAGs with the heightened intensity of violence in contemporary NIACs. Dangerous weapons such as biological and chemical weapons becoming easier to access by armed groups also create serious security concerns of abuse (Atadjanov 2011).

In this light, let us consider in more detail the likely scenarios leading to the acquisition of AWS technology and know-how, i.e., the routes and benefactors, how this is distinct from how States acquire them, and how this process may be different—or comparable—to other weapons often associated with insurgents' LOAC violations.

### 3.1. Routes and Benefactors

The first important factor impacting weapons acquisition is the availability of and access to producers or distributors. Indeed, this is a major difference between NSAGs and States, since the latter generally have more options available in terms of self-development, national procurement, or import from foreign States or producers (Pamp et al. 2018). In contrast, outside of some methods of self-obtention, which we will discuss shortly, the main suppliers of an NSAG's (sophisticated) weapons are external benefactors. Thanks to direct external support, Hezbollah was rather impressively able to arm itself with precision-guided munitions (Israel Defence Force 2019), and, even before that, obtained missiles, antitank weapons and effective weapons training from its sponsors (Erlanger and Oppel 2006). Shiites in Iraq were also reported to have received rockets, bombs, and training in explosives and sniper rifles from neighbour Iran (Grauer and Tierney 2018). During the Cold War, State sponsorship accounted for the large majority of armed groups' arsenals (United Nations 1999). While this has since somewhat decreased, States remain the most important supporters of insurgent groups, and a large number of NSAGs still rely on foreign governments as their main source of arms (Byman et al. 2001; Duquet 2009).

Having State benefactors is particularly powerful because their support is often backed by an extensive arms industry, domestic expertise and diplomatic standing. In (Pamp et al. 2018)'s view, obtaining sophisticated major weaponry—such as AWS—is extremely unlikely otherwise. As emphasised previously, effective transfers also include the transfer of know-how, and the ability to receive training from experts is one of the key advantages of having a State benefactor (Moore 2012). (Byman et al. 2001) agree that State backing can be very comprehensive, encompassing safe havens, military training, sophisticated weapons, and diplomatic backing—these "can often make a difference in helping an insurgency triumph over its government opponent". States, therefore, seem the most well placed to provide AWS support to NSAGs.

Literature also outlines a different type of benefactor, that of non-State supporting groups and individual sympathisers. In a study conducted from 1946–2004, it was found that 35 out of 111 (32%) rebel groups received support from non-State actors (Harbom and Wallensteen 2005). These can be nationals or sympathisers living abroad who share ideological or ethnic affiliations with the NSAG in question (United Nations 1999). A particular type of non-State benefactor are non-incumbent transnational ethnic kin ('TEK') (Cederman et al. 2013; Gleditsch and Salehyan 2006). TEK often take the form of large demographics seated in a different State(s) who support their ethnic brethren fighting elsewhere, such as Iraqi Kurd support for Kurds in Turkey. Palestinian, Armenian, Irish and Tamil diaspora have also been recorded as providing support to 'allied' insurgents (Byman et al. 2001). Notably, however, the support which sympathetic individuals, groups and TEK can provide is more restricted than that of States, and usually is limited to funds, political, moral and reputation support, and possibly a safe haven (if they share a border with the NSAG in question) (Hazen 2013). In any situation, sophisticated weapons and the know-how to use them will be hard to obtain from these types of benefactors (Byman et al. 2001). While it has been noted (Gleditsch and Salehyan 2006) that refugee[13] streams can sometimes be a promising avenue of obtaining arms, it is unlikely that it will allow NSAGs to reliably access sophisticated weapons of the calibre of AWS, as these persons are generally already impoverished and may even be lacking in basic resources (Byman et al. 2001).

There are three alternative routes for weapon acquisition which we may describe as 'self-acquisition'. The first takes the form of self-import, i.e., purchasing weapons from arms markets. These can be both legal and illegal, and both domestic and transborder. Often, self-import is realised through international arms dealers and black markets (Hazen 2013). Self-import became particularly important for NSAGs after many State benefactors ceased support at the end of the Cold War (Arasli 2011). For self-import, NSAGs must both maintain access to markets or brokers who supply AWS and have the necessary funds to procure the weapons. It is rather common that NSAGs rely on criminal activity or illegal trade of (natural) resources under their control to finance these purchases (ICRC 1999; United Nations 1999). Having access to a friendly international border is extremely beneficial in this respect, facilitating easy trade and weapon transport (Buhaug et al. 2009). For example, Hamas relies on the Gaza–Egypt border to obtain weapon transfers via tunnels, and Mindanao rebels were able to receive technology and know-how from Jemaah Islamiyah thanks to their coastal access (Cragin et al. 2007). As AWS also come in more mobile forms such as vehicles, drones or munitions launchers (meaning that they are not necessarily difficult to transport), they can be obtained as easily as traditional weapons through such routes. With regard to financing, however, it will likely be more challenging for a majority of NSAGs to procure sophisticated weaponry such as AWS. While advertised in *Slaughterbots* as soon being 'universally affordable', they remain high-end options. The single-use IAI Harop, for example, sold for about USD 40 million per unit in 2015 (Ahronheim 2019). They are certainly not purchasable in bulk and many NSAGs may prefer to follow the 'standard' insurgency strategy of focusing on small arms instead,[14] particularly if they have limited funds.

The second option for self-acquisition is looting from the incumbent government. For many NSAGs, looting government stockpiles was the first important push they needed to establish and expand their insurgency (Moore 2012). History shows that capturing weapons from more sophisticated opponents is a dominant strategy for asymmetric wars (Jackson 2011). In some situations where no international access is possible due to geographical restraints or embargoes—removing self-import as an option—looting may be the only source of weapons for an insurgency (SIPRI 1994). Looting generally occurs through raids at poorly guarded stockpiles (police stations, military installations or depos) (Hazen 2013), although in some corrupt countries, it is also possible to abuse weapon leaks, diverting

---

[13] The cited study uses this term in the colloquial sense, i.e., referring to groups or persons migrating for fear of harm (instead of in a legal sense, which carries legal consequences for the persons in question).

[14] See Section 3.2.

arms intended for State troops toward the armed group through bribery (Jackson 2011). (Pamp et al. 2018) mention looting as one of the few ways rebels can gain access to major conventional weapons—compared to self-acquisition through black markets, it may indeed be more likely that AWS are obtained in this way.

While promising at first glance, there are many specific prerequisites that must be fulfilled for this to become a likely route for AWS acquisition. First, looting is only possible against weak or badly organised opponents. Against stronger governments who do not leave important depos unguarded, do not abandon stockpiles in retreat,[15] and whose command structure is not vulnerable to corruption or bribery, looting and diversion becomes more difficult (Marsh 2007). Effective State control over its weapons increases barriers to looting, and dissuades groups from relying on this method. (Jackson 2011) note that a "strong State with a well-disciplined army is in a better position to protect its armoury and root out corruption, making it difficult for rebel groups to capture or buy these weapons". Secondly, "[s]tolen and looted weapons do not offer the luxury of choice" (Moore 2012), meaning that the State must have AWS in its arsenal in the first place. For the moment, AWS remain a relatively expensive option; thus, economically/militarily weak and failed States against which the looting strategy works better will also be less likely to have AWS in stock to loot. In addition, several States, such as China, Iraq, Brazil, Pakistan and Mexico (Human Rights Watch 2020), have already proclaimed support toward a prohibitory treaty on AWS. Depending on how broadly this treaty defines AWS, an opposing government which has subscribed to such an obligation will have fewer (or even no) such systems in their arsenal to loot, reducing the possibility of using this acquisition method.

The final method of self-acquisition is self-development. In 2009, for example, almost 20% of the arms (revolvers, shotguns and pistols) carried by non-State actors in Nigeria were domestically produced (Duquet 2009). At one point, the MILF in Mindanao also reportedly maintained a factory of 100 employees which produced side-arms, rifles, and even rocket-propelled grenades for their own use (Capie 2004). While this demonstrates that self-development can sometimes also produce sophisticated weaponry, this acquisition method is generally limited to small arms. In addition, self-made weapons are time-consuming to produce and frequently unreliable (Duquet 2009; Marsh 2007). Given the sophistication required for AWS, involving not just hardware production but also AI programming, it seems safe to assume that it is relatively unlikely that NSAGs can obtain AWS from this channel.

In light of the above, the most probable avenue of AWS transfer will be through State support. This route avoids the problem of limited funds, of delivery, and of obtaining the know-how required to operate the AWS. While individual sympathisers and TEK can offer financial support to NSAGs, AWS will be relatively difficult to procure compared to smaller or less advanced weapons. Major conventional arms are, in general, very hard to obtain 'in the wild' (Marsh 2019), and this will also apply to AWS. The only feasible alternative, looting AWS from the State, suffers from the irony that strong governments which are most likely to possess AWS are less likely to create the conducive circumstances which enable them to be looted. Additionally, if the opposing government has committed itself to not using AWS at all, this method is eliminated a priori. From a policy perspective, these findings suggest we should focus our attention, in particular, to States as the primary subjects to address. We return to this in a more comprehensive way in Section 5.

### 3.2. Visibility and Movement

If a group wishes to rely on a benefactor or vendor for its acquisitions, some movement needs to take place that physically transports the arms in question to its end user. The possibility of such movement—and the ease with which such transfer can be identified and restricted—is, however, not equal across all weapons. In this respect, in both literature and

---

[15] The experience of the catastrophic US retreat from Afghanistan and the subsequent looting of US weapons by the Taliban (Weaver 2021), however, shows that 'stronger governments' are not necessarily exempt from this scenario.

arms-transfer databases, a distinction is made between small arms and miscellaneous types of weapons. *Small arms* are generally understood as "weapons designed for personal use" (United Nations 1997) or "weapons that can be carried by one or two persons" (Wezeman 2002). Examples include self-loading pistols, rifles and carbines, submachine guns, assault rifles and light machine guns (United Nations 1997), but also slightly larger (but still portable) weapons such as mortars and small anti-aircraft guns (Marks 2007). Small arms are generally favoured by NSAGs for their simplicity, concealability, availability and low cost (ICRC 1999). Weapons not falling under the small-arms category are often further sub-divided into light and major conventional arms. *Light arms* are weapons designed for operation by a crew, e.g., heavy machine guns, rockets, mortars, and air defence (United Nations 1997). Tanks, armoured vehicles, fighter aircrafts, guided missiles, 100mm+ artillery, and large anti-air systems fall into the last category of *major conventional* (Holtom et al. 2012; ICRC 1999).

Because AI can be installed into any type of existing weapon system, AWS can technically come in any of these three categories. However, viewing existing weapon systems which employ AI, we can reasonably conclude that the bulk of AWS will fall into the latter two categories (light or major conventional). Currently, AI is used particularly in:

- Smart munitions and sensor-fused weapons (weapons that, after launch, search for a target signature and engage it) such as the BAE Bonus (BAE Systems 2020);
- Loitering systems (independent platforms which hover above a target area and search for a particular signature), such as the Harpy (Markoff 2014);
- Active protection and counter rocket, artillery and mortar systems, such as the Aegis and Goalkeeper (Scharre and Horowitz 2015);
- Anti-personnel perimeter defence, such as the South Korean SGR-A1 and South African Super aEgis II (Global Security 2017).

All of these systems would fall under the major conventional, or, at the minimum, light arms, category. While miniaturisation brings its own benefits (Roff 2016), as shown in the *Slaughterbots* video, the industry mostly places emphasis on *swarms* of such small platforms, which work in tandem (Thurnher 2018; UK Ministry of Defence 2011). While individually portable by one or two persons, the entire system of hundreds of such drones cannot reasonably be classified as a small arm. The possibility of low quantities of small, AI-empowered drones being launched by single insurgents notwithstanding, we can presume that the bulk of AWS that NSAGs can access will, therefore, be in the form of light or major conventional arms.

This finding has some immediate implications. While major conventional arms account for the majority of reported arms transfers (Levine et al. 1994; SIPRI 2017), they are also much more visible, and, therefore, controllable. Unlike small arms, information about major conventional transfers is, to a large extent, transparent in official and other public sources—"very few major weapon transfers are not known about at the time of transfer" (SIPRI 2017). While this does not directly prevent NSAGs from accessing the weapons per se, this heightened visibility does allow regimes already in place, such those provided by the (Arms Trade Treaty 2013), the EU (European Council 2008) or domestic law (ICRC 1999), to better control the flow of such capabilities. Indeed, one of the main problems in regulating small-arms transfers is that they are difficult to track and often made available on black markets (Levine et al. 1994; United Nations 1999; Wezeman 2002).

Small arms' accessibility is also the reason why they are generally seen as the weapon of choice of NSAGs, frequently being described as "any insurgency's defining technology" (Byman et al. 2001). Some conflicts are fought exclusively with small arms (Capie 2004). It is often argued that NSAGs, on average, actually prefer small arms over light and major conventional weapons, because small arms fit their 'playstyle'. They are ideal for stealth and asymmetric tactics (Jackson 2011), can be used by any new sympathiser or recruit without needing much training (Waszink 2011), and are easily obtainable even in the absence of a powerful benefactor or with embargoes in place (Bassiouni 2008). Some authors, such as (Van Creveld 1999), go so far as to label major conventional weapons

as entirely superfluous for NSAGs: they would be "the most useless, being either too expensive, too fast, too indiscriminate, too big, too inaccurate, or all of these".

In contrast, others take the position that the stereotype of the small-arms-loving insurgent is borne out of necessity instead of choice. In (Wezeman 2002)'s view, it is very likely that small arms are only weapons of opportunity and that many NSAGs "would probably rather have had—but could not afford—more powerful weapons to fight their wars". This seems like the more reasonable interpretation, as a lack of more sophisticated weaponry limits the military options available to an NSAG. While low-intensity insurgencies can perfectly be serviced by small arms alone, "decisive military victories are likely to be difficult without heavier weapon systems" (Hazen 2013). Particularly for pushes against government forces (which are typically better armed) or if the NSAG wishes to maintain territory for an extended period, light and major conventional arms are essential (Marsh 2007, 2019). Naturally, we are only speaking in generalisations: an NSAG's ideal arsenal is entirely dependent on its specific situation and objectives. Some NSAGs may deliberately avoid confrontations and prefer to prolong the state of insurgency because they gain more from exploiting captured resources, instead of actually fighting (Buhaug et al. 2009). For such groups, it would be accurate to say that the small-arms paradigm of warfighting would apply. For others, however, there is a clear incentive in acquiring advanced weaponry such as AWS, particularly if they wish to overcome the power disparity between themselves and the governments in power.[16]

On the one hand, therefore, there will be a strong incentive for many (but not all) NSAGs to access major conventional AWS to overcome the power disparity between themselves and government forces, particularly if these weapons are very effective and can outmanoeuvre enemy capabilities and combatants, as advertised by many AWS producers. On the other hand, such weapons are more difficult to obtain because they are less wieldy and more transparent in the global arms transfer market. This makes efforts such as arms-control agreements, blockades and embargoes potentially viable as controlling mechanisms when compared to other weapons commonly associated with LOAC violations (such as small firearms), although this will vary per individual AWS based on its characteristics.

## 4. Reasons for Violations While in Use

NSAGs are certainly not monolithic. Some groups will be more prepared to take deliberate efforts to respect applicable legal norms, while others may make violating LOAC a core aspect of their strategy (Durhin 2016; Waszink 2011). In contemporary conflicts, however, NSAGs are often disinclined to respect the customs of war. NSAGs are more predisposed to resort to unorthodox and sometimes radical strategies such as targeting civilians, torture, hostage taking, and extrajudicial executions (Bakke 2013). Many of the most serious humanitarian violations in conflicts today are committed by non-State armed actors, and some do so on a daily basis (Hamberg 2013). (Peters 1994) correctly predicted three decades ago that the future battlefield will feature not soldiers, but warriors: "Unlike soldiers, warriors do not play by our rules, do not respect treaties, and do not obey orders they do not like."

Many explanations are provided in literature of why NSAGs exhibit a high violation propensity. Indeed, such research is promoted in the context of efforts to improve LOAC compliance: "it is important to understand the reasons why violations occur so that intervention points can be identified" (Waszink 2011). In this section, we explore the likely reasons or circumstances which can cause NSAGs to utilise AWS in LOAC-noncompliant ways, since to cure or prevent, we need to understand the cause. This misuse can take any form, but will likely involve targeting-related war crimes, such as deploying weapons

---

[16] Most internal conflicts feature a government which has much higher arms sophistication compared to the NSAGs, as it had a weapons monopoly before the conflict outbreak (Pamp et al. 2018). In some situations, however, the government may be as weak as the NSAG. These conflicts are termed 'symmetric nonconventional' by (Kalyvas and Balcells 2010). Even in this case, however, the NSAG has an interest in overcoming the stalemate by obtaining and using stronger weapons, such as AWS.

unable to distinguish or in ways contrary to the principle of proportionality (Additional Protocol I 1977).[17] Chronologically, this step takes place after the acquisition process discussed in the previous section. We assume for the sake of our analysis that the NSAG already has the AWS in possession, irrespective of how difficult this process was.

Four major circumstances can be identified which could cause an NSAG to commit an LOAC violation using AWS: First, and most straightforwardly, they deliberately misuse the AWS, in contravention of applicable legal norms. Second, organisational, political and legal factors can create an environment which directly or indirectly leads to more violations by NSAGs. Third, the weapon design itself may be legally problematic, and the NSAG may not have the capacity or knowledge to alter it to a more compliant state. Fourth, technical characteristics of AWS can bring the NSAG to recklessly or negligently misuse the AWS, leading to a violation. We will discuss these possibilities in sequence. Particularly, the third and fourth factors are less immediately obvious, and we will take some time to develop these factors fully.

### 4.1. Deliberate Misuse

When we discuss reasons to *deliberately* misuse weapons, we are considering incentives or circumstances which drive NSAGs to ignore or deliberately act in contravention to LOAC. One motivator frequently offered in literature is the legal inequality suffered by NSAGs vis-à-vis the State. Contrary to captured combatants in international conflicts who become prisoners of war, LOAC does not provide for the legal protection of insurgents, and they can be tried normally under domestic law after the war ends.[18] Many (Bassiouni 2008; Sassòli 2008; Waszink 2011) posit that this is a major disincentive for compliance, because insurgents will be 'prosecuted anyway' if they lose, even if they neatly adhere to all LOAC requirements when fighting. There is also a variation in this hypothesis, namely, that the imbalance is perceived as 'unfair'—the mentality would be: 'As LOAC does not acknowledge me, I also don't have to acknowledge LOAC' (Atadjanov 2011). Finally, in some situations, the NSAG may reject the legal framework itself for ideological reasons, e.g., by labelling LOAC a 'western' imposition or invoking separate normative frameworks (such as Islamic law) (Saul 2017; Waszink 2011).

The other dominant reason for deliberate non-compliance is that not playing by the rules is, quite simply, advantageous. Frequently, compliance to LOAC is perceived as a handicap and, conversely, deliberately violating it as a type of force multiplier. In asymmetric conflicts, this force multiplier may be essential for survival. A 2010 UN report (UN Security Council 2010) notes that "armed groups have often sought to overcome their military inferiority by employing strategies that flagrantly violate international law". This is often borne out of the power imbalance characterising most NIACs, where the State has much more sophisticated armaments and is impossible to defeat 'fairly' (Reynolds 2005). NSAGs often argue that the inequality in arms between their and the government's forces "does not allow them to comply" with LOAC (Atadjanov 2011). As a result, they leverage the opponent's willingness to abide by LOAC as a positive handicap, while they themselves ignore it. (Arasli 2011) frames the accrued advantage slightly differently: "by deliberately placing themselves out of the domain of the law of the armed conflict, [the NSAG] is free to put operational considerations far ahead of humanism". Human shielding, where insurgents deliberately place civilians in close proximity to their soldiers or assets to dissuade the LOAC-abiding opponent from attacking (Dunlap 2016), is a classic example of how deliberate abuses can be militarily beneficial.

The presence of AWS in an NSAG's arsenal will change nothing of the (in the armed group's view) unfair legal framework in place for NIACs and their attitude towards it.

---

[17] Depending on the system in question, however, this can also take other forms. (McAllister 2018), for instance, discusses a hypothetical 'AI interrogator' which, in NSAG hands, can also be misused in contravention to the prohibition on torture.

[18] (Additional Protocol II 1977) only encourages (instead of obliges) States to "grant the broadest possible amnesty to persons who have participated in the armed conflict". See also (Kleffner 2007).

However, and perhaps unintuitively, there are reasons to suspect that AWS possession may correlate with a *reduced* drive to violate. One reason is that the utility of abusing LOAC to offset a power imbalance will decrease as the insurgents gain more sophisticated weaponry overall, including AWS. All other factors being equal, NSAGs that have reached the level of technological sophistication where they are able to field AWS will have a reduced need to rely on deliberate violations to gain an advantage—after all, the explicit advantage of AWS, as purported by its advocates (Defense Innovation Board 2019; Ministère des Armées France 2019; United Kingdom 2020; U.S. Air Force Office of the Chief Scientist 2015), is increased military efficiency and force protection. There is, however, also a more direct mechanism which may link AWS possession with more compliance. Assuming that the weapons' AI is properly programmed to function in accordance with LOAC and the insurgents do not take active steps to deliberately make the weapon commit violations, their deployment will be in conformity with LOAC. Indeed, this may be one of the strong utilities that AWS bring to NSAGs: the ability to procure a military advantage over the enemy while the system implements LOAC 'for' them.

If this is found to be correct, then the international community may have a reason to *encourage* the distribution of AWS to NSAGs to reduce the rate of LOAC violations overall. This position is only tenable, however, if all other risk factors as discussed below are properly addressed, as insurgents can still misuse the technology due to recklessness or ignorance, and the assumption of the technology being fully LOAC-compliant is far from a guarantee.

### 4.2. Violation-Conducive Circumstances in the NSAG

Sometimes, organisational, political or legal factors create an environment which does not foster high legal compliance. External factors can influence the group's compliance to LOAC indirectly. One common example is the situation where the insurgents have little expectation of accountability for non-compliance: when the threat of enforcement is low, there is less reason to make the effort to implement LOAC (Bassiouni 2008; Hamberg 2013). Internally, lower levels of organisation of the armed group are frequently linked with more violations. NSAGs often operate in smaller, independent units; lack a central command and control hierarchy; have little or no internal system of discipline; and may even be ideologically fragmented (Bassiouni 2008). As a result, they lack the social control structure present in standing armies to enforce compliance. A lack of internal organisation also impacts other compliance-inducing factors. It is much more difficult for such groups, for example, to provide training in LOAC to all of its members, to standardise rules of engagement, and to prevent internal disagreement in terms of how rules are to be applied and interpreted (Waszink 2011). (Saul 2017) and (Bellal 2016) both agree that a lack of dissemination of LOAC is one of the key factors driving the lack of legal compliance: many NSAG members, even at senior levels, have no or little training in LOAC, either because are simply unaware of its existence or because of a lack of dissemination. It is for this reason that reaching out to NSAGs and training them in the law is highly promoted by both scholars and practitioners, as well as official organisations such as the UN and the ICRC (Bellal 2016; UN Security Council 2010).

While a group's arsenal may have little direct relation with its internal organisational discipline, AWS acquisition could indirectly link to reduced LOAC violations, as the transfer of sophisticated arms often also comes with the transfer of organisational knowledge (especially if the benefactor is a State). While not one on one, a group's level of technological sophistication and internal organisation often correlate. (Kalyvas and Balcells 2010) operationalised sophistication this way when discussing why the overall sophistication of NSAGs dropped after the Cold War. The explanation presented is that during the Cold War, the sophistication of USSR-supported NSAGs was boosted by both material support (weapons, training, advisers) and military training, the latter of which generated the internal discipline and organisation necessary to oppose the governments they were fighting. Assuming AWS are likely to be provided by States, as discussed in Section 3.1, this support

will likely come with other types of training, including military doctrine. This positively impacts internal organisation, which, in turn, better allows the NSAG to maintain a central discipline and command structure, and implement LOAC.

*4.3. Hand-Me-Down LOAC Non-compliant Weapons*

One particularly pertinent external factor that can contribute toward NSAG non-compliance relates to the State where the AWS is *produced*. This is likely (but not always) the NSAG's benefactor. In Section 4.1, we assumed that in the absence of incentives which would induce an insurgent to modify an AWS's behaviour to deliberately abuse LOAC, the weapon itself would function in accordance with legal requirements. Of course, this is not necessarily so. It is up to the producer and the State where the weapon was manufactured to guarantee that weapons being developed are LOAC-compliant: the producer on the one hand, by programming and designing the AWS to enable use in conformity with LOAC (e.g., by ensuring controlability, predictability, performance and explainability) (Fornasier 2021; Haugh et al. 2018), and the State on the other, by placing enough safeguards on the domestic arms industry to ensure that this is carried out. The focus mainly lies with these two actors, as it is unlikely that many NSAGs will have the necessary technical and legal expertise to independently assess the legality and technical characteristics of the weapons they obtain, particularly if they are technologically complex.[19]

In this regard, LOAC places the primary responsibility with the producing State. An obligation commonly referred to as an 'Article 36 Review' instructs States to conduct evaluations of all weapons' legality under LOAC before they are placed into service. In addition, as any lawful weapon can be employed unlawfully, these reviews are also meant to delineate foreseeable effects of deployment and limitations of use to ensure compliance in targeting scenarios (ICRC 2006). While there is disagreement[20] regarding whether the obligation also legally binds non-parties to (Additional Protocol I 1977) ('API'), there is some general consensus (Lawand 2006; Schmitt and Thurnher 2013) that States are required to conduct some form of weapon evaluation before adoption (although not necessarily using the specific method mandated by API).

However, one possible danger is that the duty to review may be interpreted as tied to *adoption* and not *development*, meaning that a potential loophole can be found in cases where the weapon is *developed* in State X but not *adopted* by X's armed forces, and instead is immediately exported or transferred to an NSAG. Even though the API formulation requires review during "study, development, acquisition or adoption", with most interpretations (Daoust et al. 2002; Fry 2006; Schmitt 2017) agreeing that this should be read as requiring the weapon's legality to be assessed during each of these stages, there is still disagreement on whether 'pure exporters' must implement this obligation. While some (ICRC 2006) argue in the affirmative, many others (Blake and Imburgia 2010; Daoust et al. 2002; Parks 2005) are of the position that there is no requirement: the review obligation only attaches to weapons which are planned for adoption. Most damningly, the Commentary to API (Sandoz et al. 1987) confirms that even in the case of negative advice (i.e., the review found the weapon to be LOAC-non-compliant), there is no prohibition in API to sell or transfer. The same conclusion should be drawn for the general 'duty to review' applicable to non-API parties, as this duty is even less stringent than the API obligation.

It is also possible that the NSAG's benefactor is merely a 'transit' State, meaning that the weapon was produced in a different country before being transferred to the benefactor. The deciding factor here is similar to the previous scenario, as the duty to review attaches to adoption. If the benefactor has themselves (i.e., as a buyer) entered the AWS into their arsenal, it would have been obliged to review the weapon upon adoption (Boothby 2016;

---

[19] This is purely a logistical remark; it does not mean that NSAGs are excused from using blatantly illegal weapons merely because they claim not to have the technical or legal expertise to evaluate them upon obtention. LOAC principles governing weapon use apply equally to States as to NSAGs. See (Sivakumaran 2006; Special Court for Sierra Leone 2004).

[20] See and compare: (ICRC 2006; Jevglevskaja 2018).

ICRC 2006; Schmitt 2017), meaning that the weapons received by the NSAG can reasonably be expected to be LOAC-compliant. However, if the benefactor is purely a transit State, meaning that the goods only passed through the country before being transferred to the NSAG, the benefactor was not obliged to test the weapon for LOAC-compliance. Then, if the producer State also did not evaluate the weapon (because it regarded itself as a non-user), there is once again no guarantee that the weapon received by the NSAG is LOAC-compliant.

Thus, this risk attaches to a specific set of circumstances. If the AWS passed only through States which do not use the weapon themselves, there is no legal mechanism in LOAC that ensures that the weapon is of the requisite performance or robustness to perform its task, significantly increasing the risk of misuse.[21] Conversely, the problem is alleviated if (one of the) States where the AWS was produced or transferred through have also adopted the same weapon.

### 4.4. Technical Reasons for Reckless Misuse

The targeting process is usually an involved process for State armies, involving an intricate system of weighing objectives against different options and multiple safeguards to ensure that the chosen weapons mix is LOAC-compliant (Curtis E. Lemay Center 2019; North Atlantic Treaty Organisation 2016). During this process, officers rely on legal and technical experts to predict the weapon's effects once deployed (Thorne 2020), including potential collateral damage and higher order effects,[22] to make their determination on whether its use would be contrary to LOAC principles such as distinction and proportionality (Additional Protocol I 1977). The more complicated the weapon used, the more technical expertise is necessary to make these predictions: calculating the trajectory of a mortar is simpler than foreseeing what will happen as a GPS-guided ballistic missile is being programmed for launch. This difficulty is expounded with AWS for specific reasons inherent to its underlying technology: AI.

For one, AWS are, by nature, semi-independent systems whose AI makes many decisions autonomously once deployed. This makes at least a basic understanding of the algorithm upon which decisions are based necessary for accurately foreseeing how it will act in a real battlefield scenario (ICRC 2016). In (Vignard 2014)'s view, a "thorough understanding by the operator and the commander of the selected weapon's functions and effects, coupled with contextual information" is required for the responsible use of any weapon, particularly AWS. Even this alone could already be a difficult hurdle to reach for most NSAGs: we are reminded of the fact that securing weapons is much easier than securing know-how or experts who can provide technical instructions on a particular weapon (see Section 3). This is the case even with less sophisticated weaponry. (Jackson 2011) gives an example of the Sudan People's Liberation Army which successfully acquired major conventional arms through trade and looting, but failed to utilise them effectively "because of a lack of training and maintenance". One can expect, then, that this will be more challenging with AWS, which requires an understanding of the AI in addition to mechanical knowledge.

The impressive performance metrics of modern AI may also belie that they are not very good at adapting to unintended environments or unforeseen problems. Almost all AI today is still 'narrow', i.e., they perform extremely well in their intended task domain, but cannot extrapolate or generalise (as humans can) (Schuller 2017). There is little indication that this will change in the near future (Ministère des Armées France 2019; UK Ministry of Defence 2011). Thus, AWS received by NSAGs will also have a narrow, defined task domain for which they were designed. For example, an AI may be trained to distinguish military from civilian vehicles, but only outside of urban areas or only when it is not raining. Deploying

---

[21] If taking into consideration extra-LOAC norms, this risk can be reduced through arms control or disarmament regimes. We discuss this as a potential solution in Section 6.

[22] Collateral damage is usually calculated with the help of collateral damage estimation (CDE) software, which also requires technical expertise. See, e.g., (Chairman of the US Joint Chiefs of Staff 2012).

that system in circumstances for which it was not trained will drastically increase the risk that it will not perform as advertised. Given the nature of insurgencies, however, it is conceivable that NSAGs will not strictly adhere to these operational limitations, even if clearly defined in the technical manual attached to the weapon. NSAGs frequently lack resources and may have to deploy weapons 'creatively' to make up for the disadvantage,[23] may not recognise that the battlefield situation is beyond the AWS's capabilities, or may not possess the organisation or means to obtain the relevant intelligence necessary to make such a determination. In all of these situations, if the NSAG erroneously deploys the weapon anyway, there is a large risk of failures, which increases the possibility of non-compliance to LOAC.

An additional complicating factor concerns the type of AI used for most applications today: machine learning (as touched upon in Section 2.1). One common aspect of machine learning AI is that they produce black boxes, or 'opaque' systems (Boulanin 2016; Molnar 2019). This refers to AI of which only the input and output variables are observable, but where it is very difficult or impossible to understand the logic by which the system converts such inputs into outputs (Horizon 2020 Commission Expert Group 2020; Roy 2018). Frequently, results can also not be traced back or reverse-engineered to understand the cause of specific decisions (Independent High-Level Expert Group on Artificial Intelligence 2019; Scherer 2016). These difficulties are not experienced only by laypersons (to which insurgents will overwhelmingly belong) but even by computer scientists. It is for this reason that the industry has invested much into eXplainable AI (XAI) in an effort to 'decode' these black boxes (Arya et al. 2019; Barredo Arrieta et al. 2020). Knowing the processes by which a system reaches its conclusions is crucial for the user to develop a robust and predictive mental model of the AWS's behaviour, which in turn allows responsible use. This factor also exacerbates the previous problem of environment-dependency, as not understanding the AWS because it is opaque will prevent the insurgent from recognising that the environment in which they are deploying the system is problematic or incompatible.

## 5. Focalising Responses

Based on our findings in the previous sections, we can more clearly identify the focal points where AWS obtention by NSAGs can be influenced by policy, diplomacy, or sensibilisation. First, let us briefly summarise Sections 3 and 4 chronologically, departing from the system supplier up to its actual use by the insurgent group. We shall do this with the help of Figure 1, below, which illustrates our findings thus far in graphical form.

First, in the Transfer portion of the timeline, we found that States—both producers and distributors—are the most likely provider, with the other sources being limited in their ability to provide this type of technology to insurgent groups. As the systems are being transported to the NSAG, it is likely that a majority of them will be relatively vulnerable to what we shall call 'restrictions'—any measures that limit or block the delivery of the weapons to the NSAG, such as weapons-control treaties and embargoes (Dharia 2019; Jackson 2011)—due to their relative visibility compared to weapons such as small firearms. Turning then to the Use phase, we determined four general factors which may lead to their misuse: deliberate violations, violation-conducive environments, the uncritical use of weapons whose designs are not LOAC-compliant, and technical difficulties related to modern AI which require expertise to navigate and to which NSAGs likely will not have access.

---

23   Indeed, this is often the reason cited for why NSAGs prefer small arms in the first place. See (Jackson 2011).

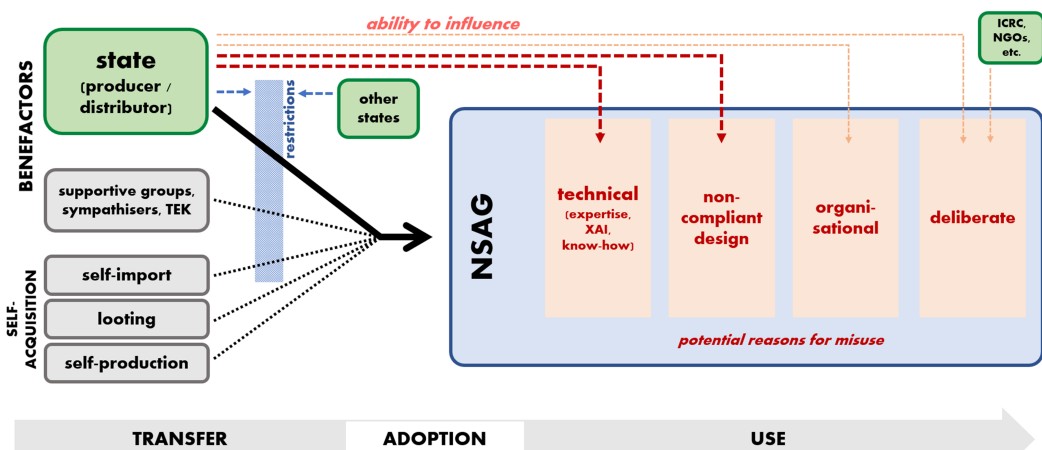

**Figure 1.** Centres of influence over potential LOAC violations by NSAGs using AWS.

More importantly, we also see that supplying States are in a unique position to exercise significant direct and indirect influence on the likelihood of NSAGs misusing AWS. The most evident way is through targeted measures to reduce deliberate misuse, e.g., by providing legal training and dissemination to the beneficiary or directly threatening to withhold support if the group does not use the delivered weapons in a lawful way. One limitation of this scheme is that while a State is required to provide legal training to its *own* armed forces (Additional Protocol I 1977), there exists no obligation in law for a supplier to also provide legal instruction to *receivers* of transfers. Its implementation is, therefore, entirely reliant on the supplier's own commitment to LOAC and its willingness to expend resources to improve the armed group's familiarity with the law. In the absence of such willingness, we would need to continue relying on independent efforts to conduct legal sensibilisation, as commonly undertaken by organisations such as the ICRC or NGOs (as indicated by the small box in the top right) (ICRC 2018b; Sassòli 2022). In contrast to this, encouraging States to provide organisational instruction does not strictly require the supplying State to be committed to LOAC, per se. Better organisation primarily leads to better cohesion, coordination and tactics, i.e., strictly military advantages. A State supplying an NSAG with AWS presumably has some strategic interest in the group succeeding, and may provide organisational instruction purely for this aim: if this indirectly leads to better compliance with the law, then this is only an added positive to be welcomed.

Above-mentioned lines of influence are not strictly related to AWS technology and are true for any type of weapons supply. In contrast, the next two (illustrated by the dark red arrows) are more specific, and should arguably become the most important focal points of future policy. The first relates to developing mechanisms that ensure, to the furthest possible extent, that AWS delivered into NSAG possession are legally compliant by design. As discussed in Section 4.1, AWS may reduce the need for NSAGs to violate the law by deploying a militarily efficient system that can implement the law 'for' them, but this only is applicable if the weapon is indeed LOAC-compliant. Ensuring such compliance is normally carried out through reviews. The main issue lies in the legal grey zone identified in Section 4.3, where the law fails to specify whether review should be conducted if the weapon is solely meant for transfer. The logic of this omission is that the law presumes that the end user will prevent the deployment of non-LOAC-compliant weapons themselves, as they *are* burdened with this review obligation upon obtaining or purchasing the weapon (ICRC 2006). However, it is far from guaranteed that NSAGs have the willingness or capacity to conduct such a review, particularly as AWS require special expertise, and perhaps novel testing methods, to properly audit (Holland Michel 2020; Meier 2019; Trusilo and Burri 2021). The realistic solution, then, is to focus our attention on the supplying States and adopt (legal or policy) measures to ensure that they—before passing the weapon to the group—are sufficiently satisfied that the weapon's baseline design is LOAC-compliant.

Review, however, only addresses the legality of weapons in abstracto, as any weapon can be employed illegally if misused (Sandoz et al. 1987). For users to be capable of utilising weapons in their arsenal correctly and lawfully, they need to receive the necessary technical training, as otherwise, they may deploy the weapon illegally (in particular, contrary to the principles of distinction and proportionality) without even intending to (Longobardo 2019). This necessity is magnified if the weapons in question are complicated ones involving modern AI, as users need to be inculcated in novel concepts such as:

- AI reliability metrics (see, e.g., (Afonja 2017; Sharma 2019));
- The fact that AI are 'narrow' and can easily fail if deployed in environments for which they were not designed or tested (Yampolskiy 2020);
- That such failures are unintuitive; and
- The fact that systems may be opaque (thus potentially necessitating training in how to use XAI solutions) (Kwik and Van Engers 2021).

Similarly to the review problem, however, there also exists no direct obligation in LOAC requiring supplying States to provide know-how when transferring weapons to other parties. In fact, contrary to legal training, this obligation does not even exist vis-à-vis their *own* armed forces (Longobardo 2019) (although some authors (Meier 2019; Roscini 2014) argue that there is an *indirect* obligation to do so, as operating weapons without technical training will likely lead to misuse). While absolutely necessary for ensuring that NSAGs do not misuse AWS, therefore, there also is no legal incentive to expend the additional effort and resources to provide technical training to beneficiaries. As with the operational component above, however, States could be motivated into providing such training for strategic incentives. Greater awareness and expertise vis-à-vis a weapon's functions and limitations will evidently lead to more precise and efficient use of resources, enabling NSAGs to militarily weaken their opponents more efficiently. Indeed, many official statements (Defense Science Board 2012; Ministère des Armées France 2019) mention the implementation of technical training programmes as a way to improve military efficiency (instead of strictly in order to comply with LOAC). As NSAG success is presumably a desired result from the supplying State's perspective, this may be used as a means to persuade the State to provide such technical training, even in the absence of legal incentives.

## 6. Closing Recommendations

The initial aim of this article was to move the discussion on AWS away from the State-centric approaches prevalent in scholarship on this topic, and toward a rarely discussed—yet highly important—actor in contemporary conflicts: the non-State armed group. As we viewed the different acquisition routes of NSAGs and the possible circumstances that could lead to AWS misuse, however, we nevertheless determined that the determinant role of States is inexorable: they remain the actors most primely positioned to influence the likelihood of LOAC violations by NSAGs. These centres of influence can roughly be found at three main positions throughout the overall acquisition process, as illustrated in Figure 1. First, as the primary (or maybe sole viable) source of the technology, they are able to influence the supply directly. Second, they can exercise influence as to how the group uses the technology. More generally, this can be achieved through legal dissemination or improving the group's organisational structure, but more specifically for AWS, this will come in two forms: (a) reviewing the AWS 'for' the group and adopting the necessary measures to ensure that the weapons transferred to the NSAG are generally IHL-compliant; and (b) providing technical empowerment for group members, which reduces the chances that the complicated and unintuitive technology is misused simply for lack of technical knowledge.

Third, States not in a supplying position, while not directly able to influence the NSAG in the same way the benefactor is, nevertheless remain principal actors in persuading these supplying States to exercise their influence on the group. The exact method employed to do so—diplomatic or more coercive—is, in this respect, not as relevant as long as it induces the State(s) in question to adopt measures such as those outlined above. For instance,

non-supplying States can invoke the principle of non-intervention (Jamnejad and Wood 2009) to exert pressure on benefactors. In *Nicaragua*, the (International Court of Justice 1986) held that the provision of weapons by the US to the Contras consituted a breach of customary law. (Ruys 2014) also considered the possibility of State responsibility for a supplying State through Art. 16 of the Draft Articles on State Responsibility (International Law Commission 2001) for aiding and abetting the war crimes committed by the recipient group, should they succeed and become the new government.

On the diplomatic front, States and the international community could consider adopting more restrictions. While prohibitory agreements (which ban transfer outright) may be effective, regulatory regimes (which merely regulate how such transfer should occur) could also suffice. For instance, the widely ratified (Arms Trade Treaty 2013) already contains provisions aimed at reducing the risk of transfers contributing toward LOAC violations. The Treaty, for instance, provides in Art. 6(3) that a State cannot transfer weapons if it has 'knowledge' that these weapons would be used by the recipient to commit international crimes (which include grave LOAC violations). It also provides that this risk must, in any way, be assessed prior to the transfer (Art. 7).[24] This is a good demonstration that restrictions can be used to reduce the risk of LOAC violations related to arms transfers. Provisions could be conceived in future agreements, for instance, which require States supplying AWS to also provide comprehensive technical training to recipients.

In addition, we identified some arguments that may even be effective vis-à-vis supplying States which are not keen on international diplomacy, nor a priori interested in the humanitarian cause of reducing LOAC violations. As we discussed, measures such as providing technical training can be justified independently from a strategic perspective, as it allows the group to fight more effectively using the weapons the supplier just provided. An added benefit is that this type of persuasion need not be conducted by States only, as groups such as the ICRC and NGOs can also employ this rationale when engaging with supplying States to ultimately achieve the goal of reducing LOAC violations.

Despite the focus in this article on NSAGs, States remain the main protagonists in efforts to reduce the risk of violations related to AWS use by NSAGs. This article provided a preliminary framework outlining points of interest that may be relevant for policymakers and LOAC advocates in the near future. Additional discussion is highly recommended on the methods and feasibility of implementing such measures, particularly in the fields of arms trade and disarmament. Existing grey zones or lacunae in the law, such as the obligation to review weapons purely meant for transfer and the provision of technical training (both to own armed forces and recipients of transfers) should be clarified and if possible expanded, as they are crucial to reduce the likelihood of AWS misuse in NIAC contexts.

As a final note, the evaluation in this paper was made on the basis of technology as available at the time of writing and some reasonable projections of near-future developments. Despite continued improvements in the affordability of high-powered processors (Scharre 2011), we are still far from a world where advanced AWS could be obtained by "anyone", as sketched in *Slaughterbots*. One possible opening for NSAGs could, however, indeed come in the form of a rise in commercial-off-the-shelf (COTS) wares. If AWS become as commercially available and as affordable as drones, for example, it will be much easier for NSAGs to obtain the technology, even in the absence of State support.[25] This would not negate this article's conclusions, but would require additional engagement with the commercial sector and related stakeholders (which may likely still include the State, e.g., its legislative body). In any event, the main points of attention would remain the same, particularly ensuring that available COTS wares are pre-reviewed for LOAC compliance

---

24 Note that by itself, this treaty would not be sufficient to mitigate all risks identified in Section 4 because the obligation in Art. 6 only triggers if there is 'knowledge' that it will be misused. In many situations we discussed, particularly related to technical reasons, there is no certainty that such misuse will occur.

25 COTS software and subsystems are already part of many military systems today, indicating a trend in that direction. See (Cummings 2018; DoD Defense Science Board 2016; Scherer 2016).

and that its 'buyers' receive some form of technical training before they can purchase the product.

**Funding:** This research received no external funding.

**Institutional Review Board Statement:** Not applicable.

**Informed Consent Statement:** Not applicable.

**Data Availability Statement:** Not applicable.

**Conflicts of Interest:** The author declares no conflict of interest.

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
