# Peer review of "Mitigating the Risk of Autonomous Weapon Misuse by Insurgent Groups"

_laws, 2022_

Round 1

Reviewer 1 Report

The Author(s) offers an interesting and insightful analysis of AWS’s proliferation to NSAGs and correctly identifies the main challenges for LOAC. The Author(s) clearly state the main assumption that the obligation of weapons review in relation to the transfer of AWS and lack of training may aggravate IHL violations committed by NSAG. They also present a novel approach to determining the main roots of the transfer of weapons to NSAGs. The arguments below would only increase the overall high quality and importance of the paper.

However, as the Author(s) notice, the paper is only an introduction to the topic, and not exhausting the whole spectrum of issues raised by AWS in the context of NSAG, for example, the question of control, nexus to an armed conflict, the necessary organizational requirement, or compliance with LOAC. These constitute the necessary determinants to consider an armed group as a party to an armed conflict, and, in relation to AWS, remain unsettled. There is a minor inconsistency in the terminology of NSAGs, insurgencies, and rebellions. I am not sure what the Author(s) mean by saying that “insurgencies generally take longer, are much messier, and involve more LOAC violations” (p. 2). Taking into account the contemporary framework of the Russian aggression against Ukraine, I am further not entirely convinced that IACs involve fewer LOAC violations than NIACs. Despite many IHL provisions being non-reciprocal, NSAG also fights for their rights, can and, in fact, comply with LOAC. It seems like the Author(s) lump together all NSAGs calling them “nefarious actors”. The crux of the problem is that the law on NIAC is generally less developed compared to IAC. The Author(s) also do not explain what specific LOAC violations concerning AWS would be committed by NSAGs. Furthermore, albeit referring to Hezbollah, the Author(s) do not distinguish between NSAG and terrorist organizations.  

I do not entirely agree that AWS is in its infancy. Technology, albeit differentiated, has already been deployed for defense purposes. It would be recommended to give real examples of AWS. The (working) definition of AWS somehow misses the lethality of autonomous weapons systems. Because of that, I have an impression that we do not discuss the question of autonomy but the question of transferring any weapons to NSAGs.

It would also be nice to provide the reader with the paper’s substantive limitations in the introductory remarks. Instead, the Author(s) explain ad hoc that something is not included in the paper. On p. 6, it is recommended to explain what the Author(s) mean by the term “refugee” since it has a strict legal consequences and follows with specific individual’s duties. While theorizing the role of powerful governments in guarding important deposits on p. 7, I would mention the case of Afghanistan and the U.S. The value of the paper would increase if the Author(s) added the obligation not to interfere with the internal affairs of another state in relation to the state’s support of NSAGs. An argument about states opting for a prohibition of AWS is not entirely true. Some indeed expressed their will to adhere to such a treaty, but only prohibiting certain (not all) AWS. It is also not clear what in p. 12 the Author(s) mean by having IHL in mind for a producer. Is it their legal obligation under LOAC? On p. 14, I do not necessarily understand what the “mental” model of AWS is. In relation to autonomy, it sounds like an oxymoron.

Indeed, the obligation of weapons review is an important preventive measure for any weapons system. However, under the AP I framework, it covers not only the legality of weapons in abstracto (p. 15) but also the circumstances in which weapons shall and shall not be legally used. This is particularly important in the context of AWS since they are not prohibited ad hoc, and therefore, targeting law applies to specifying circumstances under which AWS can and cannot be used.

Last but not least, while theorizing about the NSAG’s compliance with LOAC, the paper misses relations to the Articles on responsibility of states for internationally wrongful acts of 2001 and the eventual responsibility of a successful NSAG. 

Reviewer 2 Report

Recommend acceptance in present form subject to usual proof reading.

Author Response

Many thanks for the review.